# Minimally Invasive Spinal Treatment (MIST)—A New Concept in the Treatment of Spinal Diseases: A Narrative Review

**DOI:** 10.3390/medicina58081123

**Published:** 2022-08-18

**Authors:** Ken Ishii, Goichi Watanabe, Takashi Tomita, Takuya Nikaido, Tomohiro Hikata, Akira Shinohara, Masato Nakano, Takanori Saito, Kazuo Nakanishi, Tadatsugu Morimoto, Norihiro Isogai, Haruki Funao, Masato Tanaka, Yoshihisa Kotani, Takeshi Arizono, Masahiro Hoshino, Koji Sato

**Affiliations:** 1Department of Orthopaedic Surgery, School of Medicine, International University of Health and Welfare (IUHW), Chiba 286-8686, Japan; 2Spine and Spinal Cord Center, Department of Orthopaedic Surgery, International University of Health and Welfare (IUHW) Mita Hospital, Tokyo 108-8329, Japan; 3Department of Orthopaedic Surgery, International University of Health and Welfare (IUHW) Narita Hospital, Chiba 286-8520, Japan; 4Sapporo Spine Clinic, Sapporo 060-0042, Japan; 5Department of Orthopaedic Surgery, Aomori Prefectural Central Hospital, Aomori 030-8553, Japan; 6Department of Orthopaedic Surgery, Fukushima Medical University School of Medicine, Fukushima 960-1295, Japan; 7Department of Orthopaedic Surgery, Kitasato University Kitasato Institute Hospital, Tokyo 108-8642, Japan; 8Department of Orthopaedic Surgery, The Jikei University School of Medicine, Tokyo 105-8471, Japan; 9Department of Orthopaedic Surgery, Takaoka City Hospital, Takaoka 933-8550, Japan; 10Department of Orthopaedic Surgery, Kansai Medical University, Osaka 573-1191, Japan; 11Department of Orthopaedics, Traumatology and Spine Surgery, Kawasaki Medical School, Okayama 701-0192, Japan; 12Department of Orthopaedic Surgery, Saga University School of Medicine, Saga 849-8501, Japan; 13Department of Orthopaedic Surgery, Okayama Rosai Hospital, Okayama 702-8055, Japan; 14Department of Orthopaedic Surgery, Kansai Medical University Medical Center, Osaka 573-1010, Japan; 15Department of Orthopaedic Surgery, Kyushu Central Hospital, Fukuoka 815-0032, Japan; 16Department of Orthopaedic Surgery, Sonoda Medical Institute Tokyo Spine Center, Tokyo 121-0807, Japan; 17Department of Orthopaedic Surgery, Japanese Red Cross Aichi Medical Center Nagoya Daini Hospital, Nagoya 466-8650, Japan

**Keywords:** minimally invasive spinal treatment (MIST), minimally invasive spine surgery (MISS), minimally invasive spine stabilization (MISt), percutaneous pedicle screws (PPS), spinal instrumentation, conservative treatment, rehabilitation, preventive medicine

## Abstract

In the past two decades, minimally invasive spine surgery (MISS) techniques have been developed for spinal surgery. Historically, minimizing invasiveness in decompression surgery was initially reported as a MISS technique. In recent years, MISS techniques have also been applied for spinal stabilization techniques, which were defined as minimally invasive spine stabilization (MISt), including percutaneous pedicle screws (PPS) fixation, lateral lumbar interbody fusion, balloon kyphoplasty, percutaneous vertebroplasty, cortical bone trajectory, and cervical total disc replacement. These MISS techniques typically provide many advantages such as preservation of paraspinal musculature, less blood loss, a shorter operative time, less postoperative pain, and a lower infection rate as well as being more cost-effective compared to traditional open techniques. However, even MISS techniques are associated with several limitations including technical difficulty, training opportunities, surgical cost, equipment cost, and radiation exposure. These downsides of surgical treatments make conservative treatments more feasible option. In the future, medicine must become “minimally invasive” in the broadest sense—for all patients, conventional surgeries, medical personnel, hospital management, nursing care, and the medical economy. As a new framework for the treatment of spinal diseases, the concept of minimally invasive spinal treatment (MIST) has been proposed.

## 1. Introduction

Minimally invasive spine surgery (MISS) techniques are gaining popularity due to an aging population and the needs of patients with spinal disorders. Historically, MISS is exemplified by the transition towards minimizing invasiveness in decompression surgery. The full-endoscopic spine surgery (FESS) and microendoscopic discectomy (MED) have been spread and applied for various pathologies. During the last decade, MISS techniques have also been used for spinal stabilization techniques, which were defined as minimally invasive spine stabilization (MISt) in Japan and other Asia pacific regions in 2009. MISt procedures include percutaneous pedicle screws (PPS) fixation, lateral lumbar interbody fusion, balloon kyphoplasty, percutaneous vertebroplasty, cortical bone trajectory, and cervical total disc replacement. Thus, MISt is a MISS technique based on stabilizing the spine by implementing minimally invasive fusion and dynamic stabilization techniques for the imbalance caused by spinal instability, intervertebral instability, and spinal deformity. MISt is a widely developing field that has the potential to decrease surgical risks and improve recovery, compared with traditional open surgery. However, even MISS techniques are associated with several limitations including technical difficulty, training opportunities, surgical cost, equipment cost, and radiation exposure. These downsides of surgical treatments make conservative treatments more feasible option. As we look at the future development of medicine, medicine must become “minimally invasive” in the broadest sense—for all patients, conventional surgeries, medical personnel, hospital management, nursing care, and the medical economy. Specifically, it is necessary to re-examine conservative therapies (drug therapy, block therapy, orthotic therapy, and rehabilitation), further develop MISt techniques, develop new medical devices, reduce risks, make conventional surgery less invasive, reduce stress and radiation exposure of medical personnel, and utilize medical expenses efficiently. As a new framework for the treatment of spinal diseases, the concept of minimally invasive spinal treatment (MIST) has been proposed. The field has also seen advancements in biologics, preventive medicine, gene therapy, etc. Spinal surgeons should widely adopt the concept of MIST in their treatment.

## 2. Methods

This narrative review was based on a literature search of electronic databases PubMed and Igaku Chuo Zasshi (ICHUSHI) -Web up till July 2021. The literature was also searched, including Google Scholar. ICHUSHI-web is a bibliographic database that was established in 1903 and is being updated by the Japan Medical Abstracts Society (JAMAS), a non-profit and non-governmental body. Ichushi contains bibliographic citations and abstracts from more than 2500 biomedical journals and other serial publications published in Japan. The search strategy included the terms “minimally invasive spinal treatment (MIST)”, “minimally invasive spine surgery (MISS)”, “minimally invasive spinal stabilization (MISt)”, and “percutaneous pedicle screws (PPS)”. Based on previous literatures, we summarize the current status of MIST in the field of spine surgery.

## 3. Results and Discussion

### 3.1. History and Development of Minimally Invasive Decompression

Lumbar laminectomy was first described by Albin Smith in 1829 [1]. Nearly a century later, Mixter and Barr [2] presented groundbreaking research on the relationship between disc herniation and sciatica. Their report recommended surgical intervention to treat disc herniation and proposed a discectomy via laminectomy; however, their intradural approach necessitated a large amount of lamina and disc material to be removed. Although the Mixter and Barr technique remained the gold standard treatment for disc herniation for approximately three decades, there was still room for much improvement, including its issues involving extensive tissue dissection, excessive blood loss, and high risk of developing morbidities. To address some of these concerns, the operative microscope and the microsurgical techniques for cranial surgery developed by Yasargil [3] in the mid-1960s were later incorporated into spine surgery. The application of Yasargil’s techniques in spine surgery led to a revolution in the field that enabled smaller incisions, less blood loss, and increased visualization.

Minimally invasive spine surgery (MISS) for decompression of neural structures was introduced by Hijikata in 1975 with the use of a dilation system for percutaneous nuclectomy (PN) [4]. To provide relief from the sciatica, Hijikata described a partial discectomy and decompression of the intradiscal pressure through an anterolateral annular window. The development of these sophisticated techniques allowed safe access to the disc, introduced the effective use of guide wires, and led to the creation of a soft tissue dilating device. These techniques are now commonly applied in other subfields of neurosurgery and orthopedic surgery. The PN procedures also developed into percutaneous endoscopic discectomy (PED) and full-endoscopic spine surgery (FESS) [5,6] (Figure 1). Initially, the main indications for endoscopic spine surgery were contained disc herniation with mild canal compromise and limited migration. The development of the working channel endoscope and other associated surgical instruments aided epidural explorations and allowed the expansion of surgical indications. The expanded indications included extruded disc herniation and related conditions, such as migrated or sequestered disc herniation, recurrent disc herniation, foraminal disc herniation, and high canal compromised disc herniation.

The extraordinary advancements in endoscopic technologies have directly affected innovations in surgical techniques. The microendoscopic discectomy (MED) was first described by Foley and Smith in 1997 and allowed a minimally invasive approach for lumbar nerve root decompression [7]. Although this technique originally consisted of using an operative endoscope with the tubular system, the tubular retractors were later modified to include a microscope. The second-generation MED system called the Microscopic Endoscopic Tubular Retraction System (METRx) (Medtronic Sofamor Inc., Memphis, TN, USA) was developed in 1999, and the utilization of tubular retractors allowed for less muscle dissection, better cosmesis, decreased pain, a longer operative time, and faster postoperative recovery [8,9,10]. The MED technique has not been limited to lumbar disc herniation but has also been used for far-lateral lumbar disc herniation, lumbar canal stenosis, cervical canal stenosis, and cervical foraminal stenosis [11,12,13,14].

Hijikata’s dilation technique created a secure access route with a small incision to the affected area while avoiding tissues, and the technique remains to be the foundation of current MISS techniques. On the other hand, there were some limitations of the METRx system, including its steeper learning curve and an increased risk for nerve root injury due to limited exposure [15]. As described above, the history of MISS is exemplified by the transition towards minimizing invasiveness in decompression surgery.

### 3.2. Concept of the Minimally Invasive Spine Stabilization (MISt) Procedure

Along with the development of new instrumentation, spinal fusion techniques such as posterior lumbar interbody fusion (PLIF), posterolateral fusion (PLF), and transforaminal lumbar interbody fusion (TLIF) have advanced over the past 20 years to provide fixation and correction from the cervical spine to the ilium. There is no doubt that the therapeutic effects of these spinal fusion procedures are greatly beneficial to patients with various diseases. However, conventional spinal fusion techniques were not without its own problems, including large skin incisions and excessive bleeding due to their invasiveness, implant-related infections, and adjacent segment disease due to rigid fixation. Moreover, the rapid increase in high-risk patients with osteoporosis and pre-existing conditions in developed countries due to an aging society has created new needs to reduce the invasiveness of many conventional surgeries.

Based on the social and clinical needs of aging societies, minimally invasive spine stabilization (MISt) was proposed as a field of MISS in Japan and other Asian nations in 2009 [16]. MISt is a MISS technique based on stabilizing the spine by implementing minimally invasive fusion and dynamic stabilization techniques for the imbalance caused by spinal instability, intervertebral instability, and spinal deformity. MISt includes minimally invasive techniques as well as various techniques developed for minimizing surgical invasiveness. Furthermore, it should be noted that the term “stabilization” does not only mean the physical stabilization under surgical intervention, but also connotes the stabilization of the disease and condition. Well-known MISt techniques include minimally invasive–transforaminal lumbar interbody fusion (MIS-TLIF) [17,18], MIS-long fixation [19,20,21,22], extreme lateral interbody fusion (XLIF) [23,24], oblique lateral interbody fusion (OLIF) [25], percutaneous vertebroplasty (PVP), interspinous process motion-sparing implant, midline lumbar fusion (MIDLIF) with cortical bone trajectory (CBT) [26], and cervical artificial disc (Figure 2a).

Since 2009, MISt has become widely recognized, especially with the introduction and development of the percutaneous pedicle screw (PPS) system. It could be said that the driving force behind the popularization of MISt may be due to the wide acceptance of techniques that combines the use of PPS such as MIS-TLIF, MIS-long fixation (Figure 3), and XLIF/DLIF/OLIF (Figure 4). Today, the PPS procedure is applied for long-segment MIS spinal fixation from the cervical spine to the pelvis for pathological conditions such as spinal fractures [21], scoliosis [27], metastasis [20,22], osteoporotic vertebral fracture [28], and discitis/pyogenic spondylosis [29]. In recent years, the broadened definition of MISt includes endoscopic decompression procedures such as MED, PED, and FESS.

### 3.3. Concept of Minimally Invasive Spinal Treatment (MIST)

Due to an aging population and the needs of patients with spinal disorders, various MISt procedures have been gaining popularity, including MED, PED, FESS, MIS-TLIF, MIS-long fixation, XLIF, OLIF, PVP, interspinous process motion-sparing implant, MIDLIF with CBT, and cervical artificial disc. MISS continues to be a developing field that has the potential to decrease surgical risks and improve recovery, compared with traditional open spinal surgery. In recent years, spinal surgery has demonstrated great technological strides and innovations. The computer-assisted navigation and robotic-assisted surgery demonstrate significant advantages in complicated and intractable MIS cases. On the other hand, even MIS techniques are associated with various problems and limitations that include technical difficulty, training opportunities, surgical cost, equipment cost, radiation exposure, and risk of litigation. These problems may impose a great deal of stress on healthcare workers, place pressure on hospital management, and burden the entire healthcare economy. Although the practice of medicine must always revolve around patient-centered care, most discussions and studies on MISt procedures have narrowly focused on the procedure itself. As we look at the future development of medicine during and after the coronavirus pandemic, medicine must become “minimally invasive” in the broadest sense—for all patients, conventional surgeries, medical personnel, hospital management, nursing care, and the medical economy. Specifically, it is necessary to re-examine conservative therapies (drug therapy, block therapy, orthotic therapy, and rehabilitation), further develop MISt techniques, develop new medical devices, reduce risks, make conventional surgery less invasive, reduce stress and radiation exposure of medical personnel, and utilize medical expenses efficiently. As a new framework for the treatment of spinal diseases, we propose the concept of minimally invasive spinal treatment (MIST) (Figure 2b). The conservative Tx category includes preventive medicine such as gene therapy. Spinal surgeons should widely adopt the concept of MIST in their treatment.

### 3.4. Technical Advances in Minimally Invasive Spinal Treatment (MIST): Navigation, Augmented Reality, and Beyond

Remarkable progress has been made in minimally invasive techniques for treating the spine, including technological innovations in imaging and surgical instrumentation. Many novel minimally invasive techniques have been incorporated into clinical practice, and both spine surgeons and patients have increasingly become willing to embrace these new techniques in order to decrease the amount of tissue dissection, incision size, operation times, complications, blood loss, and length of hospitalization. Minimally invasive techniques also improve postoperative outcomes, decrease healthcare costs, and allow patients to return to daily activities earlier than conventional open techniques. As the field of spine surgery continues to develop and innovate, many novel technologies are expected to play a larger role in the field, including navigation and robotic spine surgery.

Although there are numerous advantages in MIST procedures, there are also several notable challenges such as the lack of clear anatomic landmarks and limited visualization. Navigation technology may address some of these challenges, as there has been a great amount of technological improvement in imaging resolution of the spine. Of particular note is the intraoperative three-dimensional (3D) CT-based navigation using the mobile O-arm (Medtronic, Minneapolis, MN, USA) [30]. As CT-based navigation can provide real-time virtual images, mapping of planned trajectories, and visualization of deep spine anatomy, the technology has allowed surgeons to refine their MIS techniques. The refinement of MIS techniques has led to significant reduction in instrumentation-related morbidity owing to the increased accuracy of pedicle screw and instrumentation placement in the cervical, thoracic, and lumbar spine.

The use of currently available technology in CT-based navigation can be especially useful in (1) the implantation of cages, pedicle screws, and pelvic anchors [31,32,33], (2) resection of bone tumors [33], and (3) performing the anterior floating method for the ossification of posterior longitudinal ligament (OPLL) (Figure 5a). The use of navigation is also effective for MISt with PPS, open surgeries such as corrective surgery for spinal deformity, tumor resection, and OPLL resection. There are many advantages for implementing navigation, including decreased soft tissue exposure, greater precision, and better rate of securing an appropriate resection margin. For adult spinal deformity correction, O-arm navigation allows the physician to place the minimally invasive lateral interbody cages, PPS, and S2-alar-iliac screws with precision (Figure 5b). It can be applied for 3-column osteotomies such as pedicle subtraction osteotomy and vertebral column resection to obtain clear anatomical orientation. In bone tumor resection, navigation enables surgeons to perform resections with appropriate tumor margins. The complete resection of the tumor can be performed with wide resection margin. On the other hand, several problems have also been noted for the use of navigation, including its high cost, medical exposure, interface errors involving the operator, and errors related to reference markers.

New navigation technology has led to an enormous improvement in imaging resolution of the spine. Furthermore, the use of augmented reality (AR) and mixed reality (MR) technologies have drawn considerable attention for their potential clinical applications [34,35] (Figure 6a–d). AR consists of technologies that add real-time information to moving images of the real world. AR technology is defined by the following characteristics: (1) a combination of the real and virtual, (2) real-time interaction, and (3) 3D registration [35]. AR-based navigation is a novel type of navigation that distinguishes itself from other state-of-the-art navigation systems. Philips N.V. has introduced a next-generation AR system (ClarifEye; Philips, Amsterdam, the Netherlands) equipped with a visible light camera that simultaneously captures tracking markers on the body surface via spinal fluoroscopy (Figure 6c,d). The ClarifEye system is a surface referenced navigation based on video input from four optical cameras mounted into the frame of a C-arm detector. Acquisition and patient tracking are ensured by continuous video detection of multiple markers that are randomly placed on the skin around the surgical field. In addition, instruments equipped with an optical marker (Galt Medical Corp., Garland, TX, USA) can be tracked by the navigation system in three dimensions (Figure 6c) and provide real-time feedback to the surgeon (Figure 6d).

The C-arm enables 3D cone beam CT scans (XperCT; Philips, Amsterdam, the Netherlands) for planning screw placement as well as confirming screw position. A previous study showed that PPS insertion with AR-based navigation with instrument tracking functionality is feasible and results in accuracy comparable to the standard fluoroscopy-guided percutaneous method [36]. Another study indicated that AR surgical navigation with intraoperative 3D imaging in a hybrid operating room demonstrated a statistically higher screw placement accuracy compared to the free-hand technique in a cohort of mostly spinal deformity cases [37]. Procedure time, length of hospital stays, and blood loss did not show any statistical difference between surgical techniques. In a comparative study between XperCT and O-arm, the estimated patient dose for small, medium, and large phantoms imaged by O-arm in low, standard, and high doses ranged from 9.4 to 27.6, 8.9 to 33.3, and 13.8 to 40.6 mGy, respectively. With XperCT, the estimated patient dose under the same condition were 2.8–4.6, 5.7–10.0, and 11.0–15.2 mGy. The contrast-to-noise ratio for the small, medium, and large phantoms was 2.9–3.7-, 2.0–3.0-, and 2.5–2.6-fold higher with the XperCT system, respectively [38]. AR-based navigation surgery in the field of spinal surgery has shown promising results. However, as with any new skill, this technology has a relatively steep learning curve.

For MR, the volume data of preoperative or intraoperative CT images can be used to create polygonal models to be used in Microsoft HoloLens [34]. The HoloLens can project 3D images of organs, blood vessels, and bones on the patient’s body or the actual surgical field to create an MR environment (Figure 6a,b). Devices that use MR technology can be a beneficial tool for implant placement as well as understanding the intraoperative orientation of tissues and organs. Moreover, the trajectory and location of pedicle screws can be added to polygonal data (Figure 6a,b), and the same 3D images can also be shared between multiple HoloLens for collaborative work (Figure 6a). Although this technology is still in a developmental stage, we believe that there is great potential for its use in complex spinal surgery in the future.

In the future, new technologies will continue to be incorporated into MIST. Potential fields of technology include robotics, artificial intelligence (AI), and machine learning. These fields may strengthen our preoperative planning, surgical execution, and optimal selection of patients to ensure improved postoperative outcomes and patient satisfaction. Additionally, AI technologies may be integrated into minimally invasive navigation and robotic technologies to assist spine surgeons in the accurate placement of spinal instrumentation. The field of MIST will continue to innovate and evolve over the coming years.

## 4. Conclusions

In the past two decades, various MISS techniques including endoscopic surgery and MISt procedures have offered many benefits such as small incisions, less cutting, less pain, and faster recovery after surgery. Here, we proposed the concept of minimally invasive spinal treatment (MIST), which is a new framework for the treatment of spinal diseases. In the future development of spinal treatment, treatment must become “minimally invasive” in the broadest sense—for all patients, conventional surgeries, medical personnel, hospital management, nursing care, and the medical economy. The concept of MIST includes not only MISS, but conservative therapies, further developing MISt techniques, developing new medical devices, reducing risks, making conventional surgery less invasive, reducing stress and radiation exposure of medical personnel, and utilizing medical expenses efficiently. Spinal surgeons should widely introduce the concept of MIST in their treatment.

## Figures and Tables

**Figure 1 medicina-58-01123-f001:**
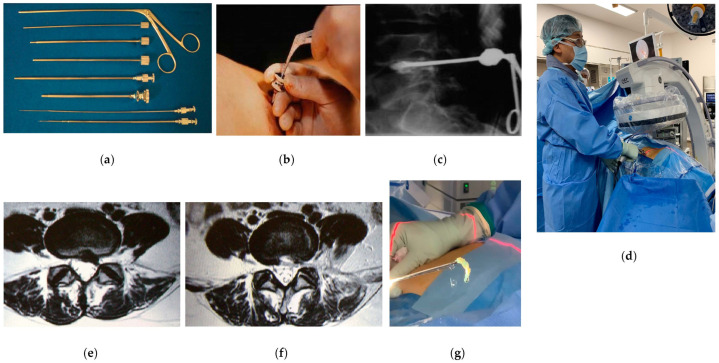
Percutaneous nuclectomy (PN) and percutaneous endoscopic discectomy (PED)/full-endoscopic spine surgery (FESS). (**a**) Dilation device of PN; (**b**,**c**) intraoperative images of PN; (**d**) intraoperative image of PED/FESS; (**e**) preoperative MR image; (**f**) postoperative MR image; (**g**) resected herniation by PED/FESS.

**Figure 2 medicina-58-01123-f002:**
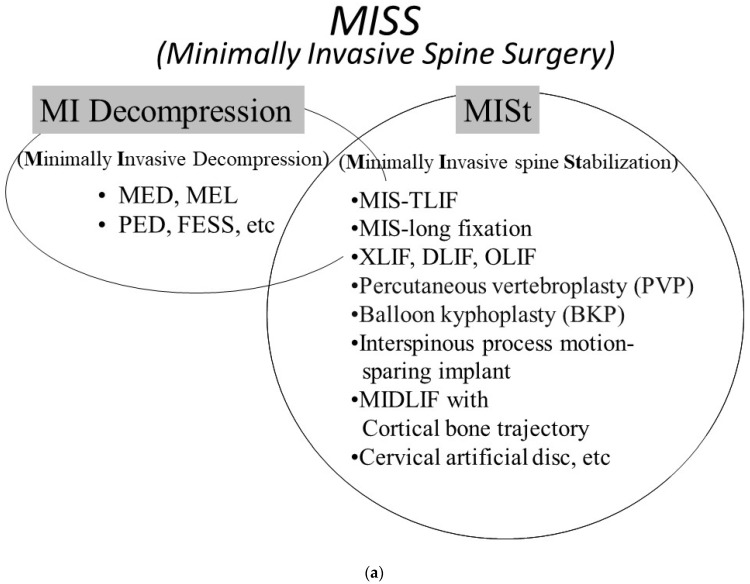
Classifications of minimally invasive therapy. (**a**) Classification of minimally invasive spine surgery (MISS). MISS is classified as two categories (MI decompression and MISt). (**b**) Classification of minimally invasive spinal treatment (MIST). MIST is classified as three categories (MI decompression, MISt, conservative and preventive therapies).

**Figure 3 medicina-58-01123-f003:**
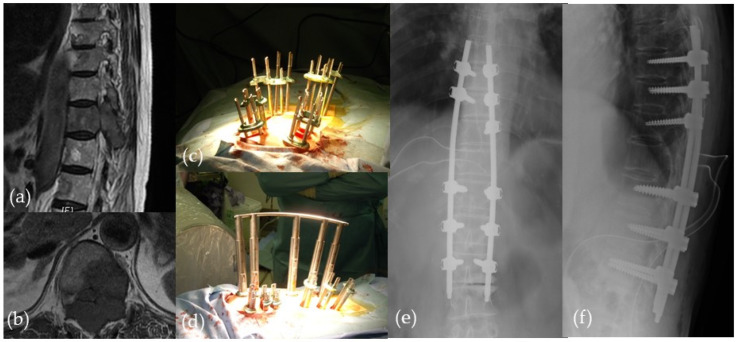
MIS-long fixation for spinal metastasis. (**a**,**b**) Preoperative MR images; (**c**,**d**) intraoperative images; (**e**,**f**) postoperative plain radiographs.

**Figure 4 medicina-58-01123-f004:**
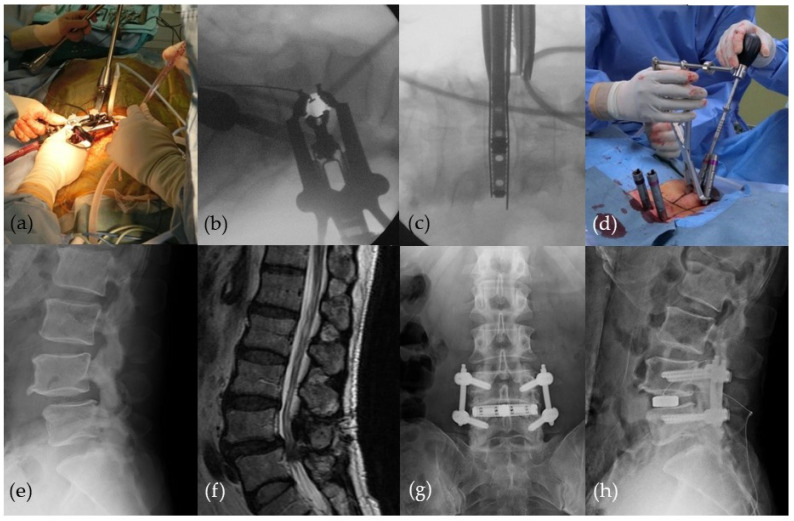
Extreme lateral interbody fusion (XLIF) combined with percutaneous pedicle screw (PPS) fixation. (**a**–**c**) Intraoperative images of XLIF; (**d**) intraoperative image of PPS; (**e**,**f**) preoperative images; (**g**,**h**) postoperative plain radiographs.

**Figure 5 medicina-58-01123-f005:**
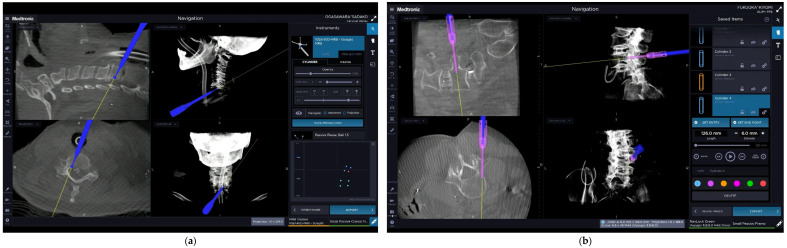
O-arm^®^ navigation monitor images. (**a**) Anterior floating surgery in cervical ossification posterior longitudinal ligament (OPLL); (**b**) percutaneous pedicle screw (PPS) insertion.

**Figure 6 medicina-58-01123-f006:**
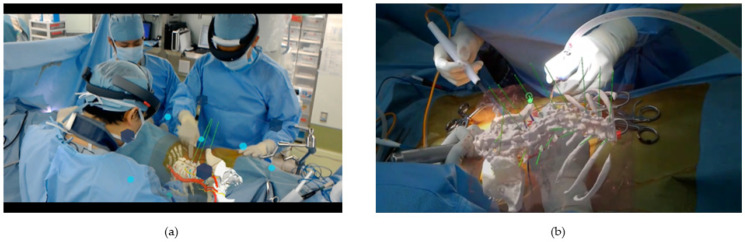
Mixed Reality (MR)-based navigation and Augmented Reality (AR) (ClarifEye^®^)-based navigation for spinal surgery. (**a**,**b**) MR-based navigation. HoloLens can project 3D images of organs, blood vessels, and bones on the patient’s body. The same projection image can be shared between the operator and assistants intraoperatively; (**c**,**d**) AR (ClarifEye^®^)-based navigation for spinal surgery. Video cameras are integrated into X-ray detector frame cover for tracking with noninvasive markers placed on the patient’s skin. Live display from the video cameras is augmented with 3D volume rendering of the spine and planned path of a pedicle screw.

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
