# Peer review of "Minimally Invasive Spinal Treatment (MIST)—A New Concept in the Treatment of Spinal Diseases: A Narrative Review"

_medicina, 2022, doi:10.3390/medicina58081123_

Round 1
Reviewer 1 Report
Thank you very much for letting me review this literature again. My previous doubts have been solved and the quality of the paper has been greatly improved. I agree to publish it
Author Response
Answer to Reviewers' comments
This manuscript was reviewed by the scientific reviewers of Medicina and many important comments have been provided. Because their comments were highly suggestive, we revised our manuscript according to their suggestions. We hope that our revised manuscript would qualify your standard and be considered acceptable.
Reviewer #1
Comment:
Thank you very much for letting me review this literature again. My previous doubts have been solved and the quality of the paper has been greatly improved. I agree to publish it
Author’s response:
We appreciate the reviewer’s comments and opinion. We hope that our manuscript gives the physicians significant impact.

Reviewer 2 Report
This topic is very current, but paper needs some revisions. Look at these points:
- It is not clear how the authors chose their papers to write this review. Can they propose a diagram flow?
- Lines 94-95: "Based on previous literatures, we summarize the current status of CAN in the field 94 of spine surgery" This part is too short. Please improve.
- There is no Results section. Please add.
- Lines 195-199: ". Today, PPS procedure is applied for long-segment IS spinal fixation from the thoracic spine to the pelvis for pathological conditions such as spinal fractures..." Improve this point, look at these ref: -- doi: 10.1097/BRS.0000000000003873. --
- Lines 301-303: " AR consists of technologies that add real-time information to moving 301 images of the real world. AR technology is defined by the following characteristics: (1) a 302 combination of the real and virtual, (2) real-time interaction, and (3) 3D registration [35]." Improve discussion about AR. Look at these refs: doi: 10.3390/ijerph19106347 -- doi: 10.1007/s00701-021-05059-9 --- doi: 10.31616/asj.2021.0437
-
Author Response
Answer to Reviewers' comments
This manuscript was reviewed by the scientific reviewers of Medicina and many important comments have been provided. Because their comments were highly suggestive, we revised our manuscript according to their suggestions. We hope that our revised manuscript would qualify your standard and be considered acceptable.
Reviewer #2
Comment #1:
This topic is very current, but paper needs some revisions. Look at these points:
- It is not clear how the authors chose their papers to write this review. Can they propose a diagram flow?
Author’s response #1:
We appreciate the reviewer’s comments. This narrative review was based on a literature search of electronic databases PubMed and Igaku Chuo Zasshi (ICHUSHI) -Web up till July 2021. The literature was also searched, including Google Scholar. ICHUSHI-web is a bibliographic database that was established in 1903 and is being updated by the Japan Medical Abstracts Society (JAMAS), a non-profit and non-governmental body. Ichushi contains bibliographic citations and abstracts from more than 2,500 biomedical journals and other serial publications published in Japan. The search strategy included the terms “Minimally Invasive Spinal Treatment (MIST)”, “Minimally Invasive Spine Surgery (MISS)”, “Minimally Invasive spinal Stabilization (MISt)”, and “Percutaneous Pedicle Screws (PPS)”. The current paper is a narrative review, but not a systematic review. Therefore, we do not provide diagram flow at this time. We hopefully submit a systematic review for Medicina special issue “MIST volume 2”. Again, we appreciate the reviewer`s comment.
Comment #2:
- Lines 94-95: "Based on previous literatures, we summarize the current status of MIST in the field 94 of spine surgery" This part is too short. Please improve.
Author’s response #2:
The sentences “This narrative review was based on a literature search of electronic databases PubMed and Igaku Chuo Zasshi (ICHUSHI) -Web up till July 2021. The literature was also searched, including Google Scholar. ICHUSHI-web is a bibliographic database that was established in 1903 and is being updated by the Japan Medical Abstracts Society (JAMAS), a non-profit and non-governmental body. Ichushi contains bibliographic citations and abstracts from more than 2,500 biomedical journals and other serial publications published in Japan. The search strategy included the terms “Minimally Invasive Spinal Treatment (MIST)”, “Minimally Invasive Spine Surgery (MISS)”, “Minimally Invasive spinal Stabilization (MISt)”, and “Percutaneous Pedicle Screws (PPS)”.” has been added.
Comment #3:
- There is no Results section. Please add.
Author’s response #3:
We appreciate the reviewer’s comments. The current paper is a narrative review, but not a systematic review. Therefore, we have provided Results and Discussion section at this time. We hopefully submit a systematic review for Medicina special issue “MIST volume 2”. Again, we appreciate the reviewer`s comment.
Comment #4:
- Lines 195-199: ". Today, PPS procedure is applied for long-segment IS spinal fixation from the thoracic spine to the pelvis for pathological conditions such as spinal fractures..." Improve this point, look at these ref: -- doi: 10.1097/BRS.0000000000003873. --
Author’s response #4:
Thank you for your helpful comment. The sentence “Today, PPS procedure is applied for long-segment MIS spinal fixation from the thoracic spine to the pelvis for pathological conditions” has been replaced to “Today, PPS procedure is applied for long-segment MIS spinal fixation from the cervical spine to the pelvis for pathological conditions”. We have also added reference entitling Navigated Minimally Invasive Posterior Cervical Pedicle Screw Fixation.
Comment #5:
- Lines 301-303: " AR consists of technologies that add real-time information to moving images of the real world. AR technology is defined by the following characteristics: (1) a combination of the real and virtual, (2) real-time interaction, and (3) 3D registration [35]." Improve discussion about AR. Look at these refs: doi: 10.3390/ijerph19106347 -- doi: 10.1007/s00701-021-05059-9 --- doi: 10.31616/asj.2021.0437
Author’s response #5:
We appreciate the reviewer’s comments. We have also added reference in discussion section.
Round 2
Reviewer 2 Report
With the exception of lines 95-104, no further changes were made to the manuscript.
Author Response
We appreciate the reviewer’s comment and opinion. We hope that our manuscript gives the physicians significant impact.
Our article is basically a narrative review (NR) entitled “Minimally Invasive Spinal Treatment (MIST)- A New Concept in the Treatment of Spinal Diseases”. The reviewer strongly requires that the article should be written according to a systematic review (SR). We have checked the definition of a NR as below.
file:///C:/Users/kenis/Dropbox/My%20PC%20(DESKTOP-RLBPNJ5)/Downloads/2047480615z2e000000000329.pdf
Here, we show excerpts from this article.
As yet there is no consensus on the standard structure of an NR. The preferred format is the IMRAD (Introduction, Methods, Results, Discussion), but an NR may be organized in a chronological order, with a summary of the history of a research when clear trends are identified, or presented as a ‘conceptual frame’, where the contents are separated according to dependent or independent variables and their relationships (Ref #1 and #2).
#1Green BN, Johnson CD, Adams A. Writing narrative literature reviews for peer-reviewed journals: secrets of the trade. J Chiropratic Medicine 2006; 5:101–117.
#2 Randolph JJ. A guide to writing the dissertation literature review. Practical Assessment, Research & Evaluation 2009;14:1–13.
Although the reviewer strongly requires IMRAD (Introduction, Methods, Results, Discussion) format according to a SR, we have written this article as a NR according to previous definition of NR. Essentially, IMRAD format including diagram flow is not required in a NR. Therefore, we have added Methods, and Results & Discussion sections except diagram flow. Otherwise, we can delete Methods and Results sections. Thank you for your understanding. Again, we appreciate the reviewer`s comment.
This manuscript is a resubmission of an earlier submission. The following is a list of the peer review reports and author responses from that submission.
Round 1
Reviewer 1 Report
This is a good-writing review, concerning on MIST. This manuscript demonstrated an excellent writing skills and considerable materials of graphs, which reflected authors huge workload. Most noteworthy is the description of cutting-edge technologies in the field of MIST. I believe that researchers and surgerons will be intrigued by this.
I’ve got an opinion, in the section of technical advances,a specific case might be reported to vividly demonstrate the novel technology of MIST. Overall this is a good job producing a publishable paper appropriate for the quality and contents of this journal. I can recommend its acceptance in the current form right away. Good job to you all and a happy new year!
Reviewer 2 Report
This topic is very current, but please look at these points:
- It is not clear how the authors chose this papers to write this review. Can they propose a diagram flow?
- Although it is a review, a better organization in materials/methods and results would be appropriate. Can authors reorganize this paper?
- In Figure 2 it is not clear whether figures 2a and 2b are sequential or opposite. Please explain this better.
- Lines 166-170: ". Today, PPS procedure is applied for long-segment IS spinal fixation from the thoracic spine to the pelvis for pathological conditions such as spinal fractures..." Improve this point, look at these ref: -- Establishing the Injury Severity of Subaxial Cervical Spine Trauma: Validating the Hierarchical Nature of the AO Spine Subaxial Cervical Spine Injury Classification System. Spine (Phila Pa 1976). 2021 May 15;46(10):649-657. doi: 10.1097/BRS.0000000000003873.
- Conclusion must be improved. What does this review add new to the literature?